# Botulinum Toxin Therapy: A Series of Clinical Studies on Patients with Spasmodic Dysphonia in Japan

**DOI:** 10.3390/toxins13120840

**Published:** 2021-11-25

**Authors:** Masamitsu Hyodo, Kento Asano, Asuka Nagao, Kahori Hirose, Maya Nakahira, Saori Yanagida, Noriko Nishizawa

**Affiliations:** 1Department of Otolaryngology-Head and Neck Surgery, Kochi Medical School, Nankoku 783-8505, Japan; nagaoa@kochi-u.ac.jp (A.N.); hiroseka@kochi-u.ac.jp (K.H.); 2Integrated Center for Advanced Medical Technologies, Kochi Medical School Hospital, Nankoku 783-8505, Japan; k-asano@dmi.med.osaka-u.ac.jp; 3Department of Medical Innovation, Osaka University Hospital, Suita 565-0871, Japan; 4Rehabilitation Department, Kochi Medical School Hospital, Nankoku 783-8505, Japan; jm-nakahira-m@kochi-u.ac.jp; 5Department of Communication Disorders, School of Rehabilitation, Health Sciences University of Hokkaido, Tobetsu 061-0293, Japan; s.yanagi@hoku-iryo-u.ac.jp (S.Y.); nisizawa@hoku-iryo-u.ac.jp (N.N.)

**Keywords:** spasmodic dysphonia, nationwide survey, diagnostic criteria, severity grading, placebo-controlled double-blind clinical trial

## Abstract

Spasmodic dysphonia (SD) is a rare voice disorder caused by involuntary and intermittent spasms of the laryngeal muscles. Both diagnosis and treatment have been controversial. Therefore, a series of clinical studies has recently been conducted in Japan. A nationwide epidemiological survey revealed that adductor SD predominated (90–95% of all cases; 3.5–7.0/100,000), principally among young women in their 20s and 30s. To facilitate early diagnosis, we created diagnostic criteria for SD and a severity grading system. The diagnostic criteria include the principal and accompanying symptoms, clinical findings during phonation, the treatment response, and the differential diagnoses. The severity grade is determined using a combination of subjective and objective assessments. Botulinum toxin (BT) injection is the treatment of choice; however, there have been few high-quality clinical studies and BT has been used off-label. We conducted a placebo-controlled, randomized, double-blinded clinical trial of BT therapy; this was effective and safe. BT treatment is now funded by the Japanese medical insurance scheme. Studies thus far have facilitated early diagnosis and appropriate therapy; they have fostered patient awareness of SD.

## 1. Introduction

Spasmodic dysphonia (SD) is a rare form of focal laryngeal dystonia characterized by involuntary and intermittent spasms of the intrinsic laryngeal muscles [1,2]. According to the muscles involved, there are three forms of SD: adductor, abductor, and mixed. Adductor SD (ADSD) is associated with strained, strangled, and effortful voice quality because intermittent tight glottal closure blocks glottal airflow [2,3]. Abductor SD (ABSD) is associated with intermittent breathiness, pitch alterations, and aphonia, which are caused by spasmodic glottal opening during phonation. Mixed SD features a combination of both voice characteristics. ADSD constitutes approximately 85–95% of all SD, while ABSD constitutes 5–10% [4,5,6]. Mixed SD is rare. Considering the impairment of smooth conversation, patients encounter substantial problems in work and daily life.

Although no consensus has emerged regarding SD etiology, the combination of a genetic factor and environmental modifiers may trigger disease development [7]. Three physiological mechanisms may be involved: decreased inhibition, increased plasticity, and abnormal sensory input to the central nervous system. The emerging picture of SD is of a disordered inhibition in response to sensory feedback during phonation [3].

Treatment of SD includes botulinum toxin (BT) injection into the laryngeal muscles, as well as surgical procedures and voice therapy. The most common and effective treatment is BT injection, but BT is used off-label in most countries [6]. BT inhibits acetylcholine release at the neuromuscular junction, reducing muscle activity. The target muscles are the thyroarytenoid in ADSD patients and the posterior cricoarytenoid in ABSD patients. Surgical interventions include type 2 thyroplasty [7,8], thyroarytenoid myectomy [9,10], and selective laryngeal denervation-reinnervation [11]. These interventions are indicated for ADSD, as they reduce tight glottal closure. Voice therapy also seeks to reduce forceful glottal closure; however, its efficacy is limited.

Despite recent advances in our understanding of SD, the epidemiological data have been inadequate because of large survey absence; moreover, diagnostic criteria have not been established and BT use has been off-label. Thus, several clinical research projects were recently performed in Japan. These greatly increased our knowledge of SD and yielded valuable therapeutic evidence.

## 2. Epidemiological Surveys of SD

### 2.1. Previous Studies Worldwide

Only a few SD epidemiological studies have been reported thus far (Table 1). Nutt et al. [12] conducted population-based epidemiological research in Rochester, NY (USA); they reported an SD prevalence as 5.2/100,000 (95% CI: 1.1–15.1). Although the study region was limited, the data appear to be reliable because the medical records of all individuals in the city were accessed via the Rochester Epidemiology Project [6].

Duffey et al. [13] and the Epidemiological Study of Dystonia in Europe Collaborative Group [14] reported a lower prevalence (0.8 (0.5–1.3)) in North England and a prevalence of 0.7 (0.5–0.9) in eight European countries. Konkiewitz et al. [15] reported that the SD prevalence in Munich was 1.0/100,000 (0.4–1.5) based on the number of SD patients among individuals who visited two university hospitals for BT therapy to treat dystonia. The study area was limited, and patients who did not receive BT therapy may have not been included [6]. Pekmezović et al. [16] collected data concerning idiopathic focal dystonia in Belgrade, Yugoslavia; the prevalence of laryngeal dystonia was 1.1/100,000. In a social health service-based survey, Asgeirsson et al. [17] reported that the prevalence of primary laryngeal dystonia in Iceland (population approximately 288,000) was 5.9/100,000. This was the first nationwide SD survey worldwide. The National Spasmodic Dystonia Association of the USA estimated that about 50,000 individuals exhibited SD in North America [18]. This equates to a prevalence of 13.7/100,000. However, this number may be inaccurate because of (ongoing) misdiagnosis or missed diagnoses. In this literature, the male:female ratios were 1:1.1 to 1:2.4 and the age of onset ranged from 35 to 50 years.

### 2.2. Epidemiological Surveys in Japan

In Japan, Yamazaki [19] performed a questionnaire survey of 81 university hospitals in 2001. This identified 169 patients, and showed that 90.5% had ADSD and 9.5% had ABSD. The male:female ratio was 1:4.4 and the mean patient age was 36.7 years; the SD prevalence was 0.94/100,000 [19]. However, only university hospitals were surveyed; therefore, the data may be unreliable. Yanagida et al. [20] performed a survey in Hokkaido (the northernmost island of Japan, with a population of 5.4 million) and identified 85 patients during 4 years of the study period; 89.4% had ADSD, 8.2% had ABSD, and 2.4% had mixed SD. The male:female ratio was 1:4.3 and the mean age was 32 years. The most common age of onset was 20–29 years (44.7%).

To obtain updated and detailed clinical data, we conducted a nationwide Japanese survey in 2015 [6]. This identified 1534 patients treated over 2 years. ADSD predominated at 93.2%, followed by ABSD (5.7%) and mixed SD (1.0%). The male:female ratio was 1:4.1 and the mean patient age was 38.9 years (Figure 1). Over half of all patients were in their 20s or 30s. The mean age of onset was 30.9 years, and the most common decade of onset was the 20s (36.9%). The interval between disease onset and SD diagnosis ranged from 1 month to 46 years with a median of 3.0 years. Over 20% of patients were not diagnosed within 10 years of onset (Figure 2). These data caused us to speculate that the prevalence of SD was 3.5–7.0/100,000 in Japan. BT injection was the most common treatment (41.0%) for ADSD; however, this was performed off-label in only a few institutions, thus imposing time and financial costs on patients. Surgical treatment was less common (24.7% of treatments) but was more preferred than at the time of the Yamazaki report [19]. Type 2 thyroplasty using titanium bridges [8] was most often performed.

### 2.3. Epidemiological Characteristics of SD

The prevalence among the reports varied greatly. This may be due to differences in epidemiological survey methods and lack of diagnostic criteria of SD. Although an additional large-scale survey is needed, the prevalence of SD in Japan is similar to the prevalence in Rochester [12] and Iceland [17]. The high predominance of ADSD (90−95%) is consistent with previous reports (Table 1). Women exhibit four-fold greater risk, compared with men. This sex difference is larger than in the USA (1:1.7) (Blitzer et al. [4]), Australia (1:1.6) (Tisch et al. [21]), and other countries. The high prevalence in Japanese women may reflect a racial difference or a socio-medical concern; perhaps women are more concerned about voice disorders, compared with men. In our survey, 60% of patients were in their 20s or 30s and the mean age of onset was 31 years, similar to the findings in earlier Japanese reports [19,20]. The mean age of onset was 45–56 years in other countries [15,16,17,21,22,23]. Only Nett et al. [12] reported an age of onset similar to the age in Japan. Japanese patients appear to be younger than patients in other countries. The median disease duration from onset to diagnosis was 3.0 years in our report. Creighton et al. [23] reported that a mean interval of 4.4 years elapsed prior to SD diagnosis. Both physicians and patients find diagnosis difficult and lack disease awareness.

## 3. Creation of the Diagnostic Criteria and a Severity Grading System

### 3.1. Diagnostic Criteria

Because SD is rare, most physicians never see SD patients. Furthermore, there is no specific examination; SD diagnosis is difficult and often delayed. We thus created diagnostic criteria based on the clinical characteristics (Table 2) [6]. We consider the main and accompanying symptoms, clinical findings during phonation, the treatment response, and the differential diagnoses. The principal symptoms are four in number for ADSD and ABSD, each. The accompanying voice symptoms aid diagnosis, although they are not specific for SD. The clinical findings include intermittent movements of the vocal folds observed on laryngeal endoscopy, abnormal laryngeal movement or cervical posture, and reduction of vocal symptoms using a sensory trick during phonation. The effectiveness of BT injection and insufficient improvement of voice symptoms after voice therapy are supportive findings. The differential diagnoses include voice tremor, muscle tension dysphonia, psychogenic dysphonia, and stuttering. Definite and possible SD are diagnosed depending on the number of items that meet the criteria [6].

In terms of diagnostic tools, Ludlow et al. [24] proposed a three-step procedure that consists of a screening questionnaire (possible SD), clinical speech examination (probable SD), and laryngeal examination (definite SD) [24]. The procedure is concise and clinically useful, although it requires further validation. Our criteria are generally consistent with their approach and convenient for use in the clinic.

### 3.2. Severity Grading

In general, disease treatment is determined by disease severity and individual characteristics. Several parameters have been used to evaluate the severity of SD, including the Voice Handicap Index (VHI) [25,26,27], the Voice-related Quality-of-Life [25,27,28], auditory-perceptual dysphonia severity [29,30], the durations of phonatory breaks [31,32], and the presence of aperiodic segments [31,32]. Our severity grading system evaluates both subjective and objective disorders in phonation or speech (Table 3). We define subjective and objective subcategories when grading. Subjective assessment features the VHI and evaluation of socio-psychological impairment. The scores range from 0 to 3. Objective severity is scored from 0 to 3 by physicians who listen to free talk or recitations of standardized sentences. The overall severity grade is a combination of the subjective and objective scores: mild, moderate, or severe. The VHI is a widely used self-assessment instrument (see below). The VHI reflects the extent of daily life impairment imparted by dysphonia; many recent studies have used the VHI to measure SD severity [25,26,27].

### 3.3. Significance of the Diagnostic Criteria and Severity Grading

Diagnosis of SD has been difficult; a suitable clinical tool has not been available. SD has been misdiagnosed as psychogenic dysphonia or other functional phonation disorders. Many years may pass before diagnosis [6,23]; patients are not treated, become upset, and subsequently become depressed. Our diagnostic criteria and severity grading can facilitate early diagnosis and prompt treatment. Furthermore, they educate clinicians and patients. The tools are provisional; we are currently verifying reliability and validity.

## 4. Clinical Trial of BT Therapy

BT injection is generally accepted as the treatment of choice for SD [2,33,34,35]. However, few well-controlled clinical studies have been performed [36]. Truong et al. [37] conducted a prospective, randomized, double-blinded clinical study with blinded outcome assessments. The BT group exhibited significant reductions in voice perturbation and the fundamental pitch range; spectrographic voice characteristics and speech scores improved in that group. This has been the only high-quality study of BT therapy; however, the parameters measured are not specific for SD. Moreover, BT for SD was not previously approved in any North American, European, or Asian country [33,38]. Therefore, we performed a multi-center, placebo-controlled, randomized, double-blinded parallel-group comparison/open-label clinical trial of BT (Botox) therapy (the “BOtulinum toxin Injection therapy for Spasmodic dySphonia” [BOISS] study) [38]. Based on the results, BT for SD was approved in Japan in 2018.

### 4.1. Study Design

The BOISS study was performed as an investigator-initiated clinical trial that followed Good Clinical Practice guidelines; a multi-center study was conducted at eight Japanese institutes [39]. The inclusion criteria were age ≥ 12 years, duration of SD voice symptoms ≥ 6 months, and moderate-to-severe SD. The exclusion criteria were concurrent systemic neuromuscular disease excluding dystonia, vocal fold paralysis or a symptomatic swallowing disorder, prior surgical treatment for SD, and voice therapy within the previous 8 weeks or BT therapy within the previous 24 weeks [38]. The initial injections into ADSD patients followed a placebo-controlled, double-blinded randomized procedure; up to two re-injections were permitted as an open-label study. The drug or placebo dissolved in saline was injected into the thyroarytenoid muscle trans-cutaneously through the cricothyroid membrane under electromyography guidance. The initial injection dose was 2.5 U and the re-injection dose was 1.0–2.5 U, depending on the initial response [38]. For ABSD patients, each injection was open-label, because ABSD is extremely rare and double-blinding was not feasible. The drug was injected unilaterally into the posterior cricoarytenoid muscle by an anterolateral transcervical approach under electromyography guidance. The initial injection dose was 5 U and re-injection doses were 2.0–5 U [38]. For both SD types, the intervals between injections were ≥12 weeks. After each injection, patients were followed up at 2 and 4 weeks, then every 4 weeks thereafter, for the entire 48-week study period. At each visit, the number of aberrant morae, the GRBAS scale, VHI, and visual analog scale (VAS) scores were collected. Trans-nasal laryngeal endoscopy was performed; phonatory function and blood chemistry were examined. The primary endpoint was the change in the number of aberrant morae at 4 weeks after drug administration. The secondary endpoints included changes in the number of aberrant morae, the GRBAS scale score, the VHI score, and the VAS dysphonia severity score during the entire study period [38].

### 4.2. Methods for Evaluation

#### 4.2.1. The Mora Method

In Japanese, mora indicates a minimum rhythmic sound unit (a phoneme) represented by a single vowel or consonant-vowel complex. Japanese words are composed of morae, analogous to the syllables of English. The phonatory disorders of SD are characterized by aberrant mora production. According to the method of Kumada et al. [39,40], we asked ADSD patients to read aloud a specific sentence consisting of 25 morae with many vowels and voiced consonants that SD patients find difficult to pronounce. Similarly, ABSD patients read a sentence consisting of 27 morae with many voiceless consonants. All voices were digitally recorded and three phonetics experts counted the numbers of aberrant morae separately [38]. The median values were used in analysis. Patients with initial aberrant morae values of ≥12/25 and ≥5/27 (ADSD and ABSD, respectively) were enrolled in the present study.

#### 4.2.2. GRBAS Scale

The GRBAS scale objectively assesses auditory/perceptual voice quality by exploring the grade (G) of hoarseness, roughness (R), breathiness (B), asthenia (A), and strain (S), each of which is scored as 0, normal; 1, slight; 2, medium; or 3, high [41,42]. The voice disorders of SD are characterized by abnormally high scores for (S) in ADSD and (B) and/or (A) in ABSD. In our study, the GRBAS scale was assessed by each physician.

#### 4.2.3. Voice Handicap Index

The VHI is a patient-rated scale developed by Jacobson et al. that explores the severity of a disability caused by disturbed verbal communication [43]. It includes 30 items divided into functional, emotional, and physical domains (10 each). Each item is scored on a 5-point scale as follows: 0, never; 1, almost never; 2, sometimes; 3, almost always; and 4, always. The total scores range from 0 to 120; more severe subjective voice disorder is indicated by a higher score. We used the Japanese version of the VHI with minor modifications [44].

#### 4.2.4. Visual Analog Scale

Participants subjectively assessed dysphonia severity using a 100-mm VAS; higher scores indicated that phonation was more affected by SD. The left and right anchor points corresponded to no dysphonia and the worst possible dysphonia, respectively. An assessor recorded all scores.

### 4.3. Results

#### 4.3.1. Treatment Effects and Clinical Courses

Twenty-two ADSD patients and two ABSD patients were enrolled. Of the ADSD patients, 11 were assigned to each of the BT and placebo groups. The changes in aberrant mora numbers at 4 weeks after injection (primary endpoint) were −7.0 ± 2.30 and −0.2 ± 0.46 in the BT and placebo groups, respectively (Table 4) [38]. The least mean-squares difference (95% confidence interval) between the two groups (−6.5 (−11.6, −1.4)) was statistically significant. Compared with baseline, the numbers of aberrant morae significantly decreased during 2–12 weeks post-injection, with a peak at 2 weeks. The (S) element of GRBAS also decreased after 2 to 8 weeks. The VHI score significantly decreased from 4 to 12 weeks from the baseline, but it did not show a significant change at 2 weeks. Comparison between BT and placebo groups showed insignificance. The VAS score revealed some post-injection improvement, but the difference was not statistically significant [38,45]. Laryngeal endoscopic examination commonly showed partial paralysis and mild bowing of the vocal fold in the treated side at 2 weeks post-injection and recovery from these conditions at 4 weeks. The placebo group exhibited no change at any timepoint. Younger (<40 years, *n* = 12) patients tended to exhibit larger changes in all parameters, compared with older (≥40 years, *n* = 10) patients at 4 weeks after treatment, but the differences were not statistically significant. Nineteen patients received at least two BT injections. We compared the changes in parameters at 4 weeks after the initial and second injections. The (S) element of GRBAS exhibited a significantly greater improvement after the second injection than after the initial injection [45]. No significant differences were apparent in aberrant mora number, VHI score, or VAS score. In the two ABSD patients, one showed no improvement in the number of aberrant morae after injection. In the other patient, a moderate decrease was observed; the change after BT re-injection was greater than the change after initial injection [38]. The G and S GRBAS scores also improved 2 and 4 weeks after the injection. No significant changes in VHI or VAS scores were apparent.

#### 4.3.2. Adverse Events (AEs)

In ADSD patients, the most frequent AE was a voice disorder (77.3%), followed by a swallowing disorder (40.9%), during the entire study period [38]. In the BT group, voice and swallowing disorders after the initial treatment were seen in 63.6% and 36.4%, respectively. In contrast, they were seen in 18.2% and 0% in the placebo group. The voice and swallowing disorders were characterized by breathy hoarseness and liquid aspiration, respectively. These AEs were mild and resolved after 25.8 and 15.8 days, respectively. One ABSD patient showed a mild voice disorder after initial BT injection, but it resolved within 4 days [38].

#### 4.3.3. Factors Influencing the Treatment Effect

Our clinical trial is the first high-quality study (based on Good Clinical Practice guidelines) of BT therapy for SD. We measured changes in the numbers of aberrant morae; these are specific for SD. Noteworthy, the objective parameters (number of aberrant morae and GRBAS score) exhibited peaks of improvement at 2 weeks, but the subjective parameters (the VHI and VAS) exhibited peaks of improvement at 4 weeks. This finding can be explained as follows: BT injection triggered breathy hoarseness resulting in unsatisfactory subjective voice improvement at 2 weeks, but the hoarseness gradually disappeared within 4 weeks, accompanied by subjective voice improvement [45]. Notably, repeat BT injection tended to increase the therapeutic efficacy. Many ADSD patients exhibited compensatory supraglottic hyperadduction, which mitigated the effects of BT injection. Repeated BT injection could gradually reduce effortful phonation, thus enhancing therapeutic efficacy. Younger patients tended to respond better, compared with older patients. We speculated that younger patients exhibit greater plasticity of the disordered central nervous system [45].

After the initial BT injection, 12 ADSD patients presented with breathy hoarseness and/or aspiration of drinking liquids. Ten patients reported neither symptom. Changes in the parameters, number of aberrant morae, and the (S), VHI, and VAS scores, at 4 weeks after injection in patients with AEs were significantly greater than those in patients without AEs [46]. Previous studies also found that patients with hoarseness or aspiration after the treatment experienced better or longer treatment effects [21,35,46]. Thus, these AEs are presumed to mirror the therapeutic effects.

### 4.4. Role for BT Injection Therapy

We confirmed the efficacy and safety of BT therapy for SD. The therapy is less invasive than surgery and it is highly efficacious. Therefore, BT injection is a first-line treatment. Figure 3 shows a treatment flow diagram. As a first step, voice therapy should be applied to reduce secondary acquired effortful phonation manner. BT injection can be used for any severity grade of SD. Disadvantage of BT therapy is that it is not a definitive treatment and repeat injections are essential. This forces the patients to undergo long-term treatment and necessitate high treatment cost. On the other hand, surgical intervention is expected to have long-lasting therapeutic effect and is indicated to the patients with moderate to severe ADSD. If the symptom recurs of insufficiently improves after the surgery, supplementary BT injection can be performed. For ABSD, effective treatment procedure is not established, therefore, BT injection can only be expected to improve the symptoms. Multiple treatment options are currently available.

## 5. Conclusions

SD is rare; both diagnosis and treatment have been challenging. We thus conducted a series of clinical studies. A nationwide epidemiological survey revealed the prevalence and clinical features of SD in Japan. Our creation of diagnostic criteria and a severity grading system facilitate early diagnosis and treatment. We performed a placebo-controlled, randomized, double-blinded clinical trial of BT injection therapy; we demonstrated that this therapy was effective and safe. The Japanese national medical insurance system then approved the use of BT to treat SD. We hope that our work encourages establishments of similar diagnostic criteria and treatment strategy in other parts of the world to aid both patients and physicians.

## Figures and Tables

**Figure 1 toxins-13-00840-f001:**
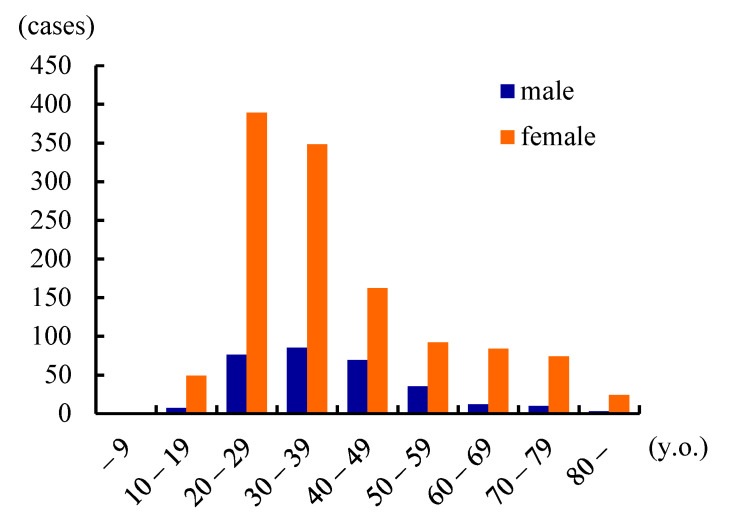
Age and gender of SD patients. Most patients are aged 20–39 years, and females are four-fold more than males. (Reproduced with permission from Hyodo M, Auris Nasus Larynx, published by Elsevier, 2021. [6]).

**Figure 2 toxins-13-00840-f002:**
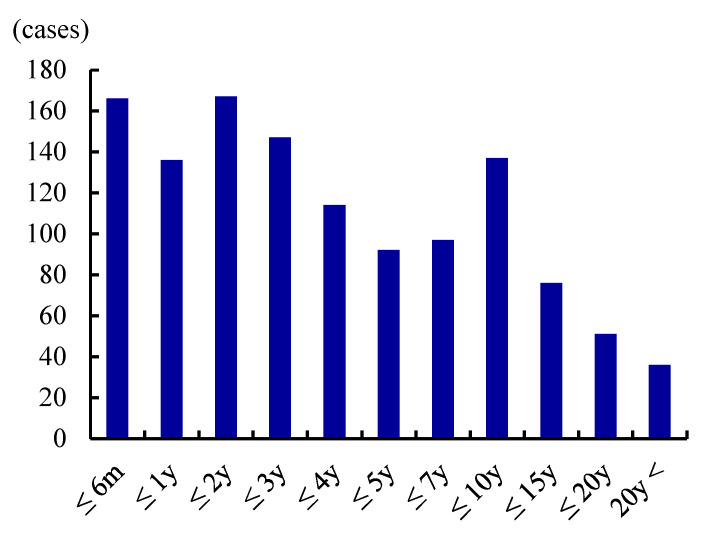
Duration from onset to diagnosis. In half of patients, the diagnosis was delayed for more than 3 years. (Reproduced with permission from Hyodo M, Auris Nasus Larynx, published by Elsevier, 2021. [6]).

**Figure 3 toxins-13-00840-f003:**
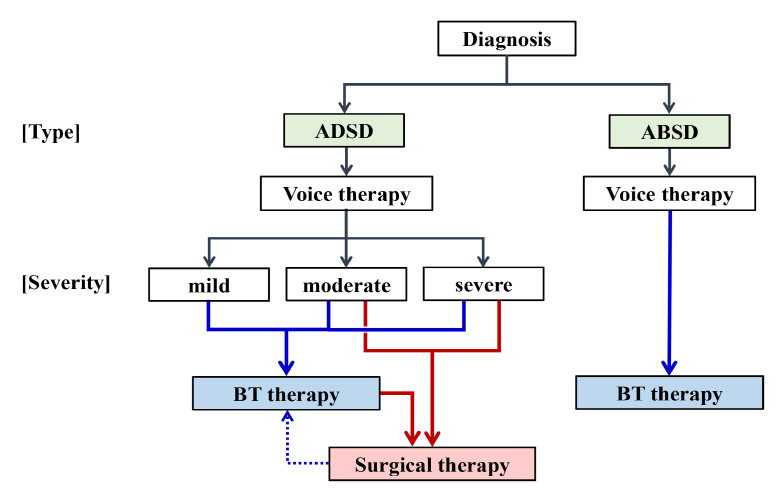
A proposed treatment flow for SD. Botulinum toxin (BT) injection therapy is indicated for any severity grade of ADSD and ABSD.

**Table 1 toxins-13-00840-t001:** Literature review of epidemiological studies for spasmodic dysphonia. (Reproduced with permission from Hyodo M, Auris Nasus Larynx, published by Elsevier, 2021. [6]).

Publications	Prevalence (/100,000)	Male:Female	Age of Onset (y.o.)
Nutt, et al. (Rochester, USA) (1988)	5.2 (1.1–15.1)	1:1	35
Duffey, et al. (Northern England) (1998)	0.8 (0.5–1.3)	N/A	N/A
ESDE (Europe) (2000)	0.7 (0.5–0.9)	N/A	N/A
Castelon Konkiewitz, et al. (Munich, Germany) (2002)	1.0 (0.4–1.5)	1:1.3	48.0
Pekmezović, et al. (Belgrade, Yugoslavia) (2003)	1.1 (0.6–1.9)	1:1.6	46.3
Asgeirsson, et al. (Iceland) (2006)	5.9 (3.4–9.4)	1:2.4	50.1
NSDA (North America) (2019)	13.7	N/A	N/A
Yamazaki (Japan) (2001)	0.9	1:4.4	36.7
Yanagida, et al. (Hokkaido, Japan) (2016)	1.6	1:4.3	32
Hyodo, et al. (Japan) (2016)	3.5–7.0	1:4.1	30.9

ESDE: The Epidemiological Study of Dystonia in Europe Collaborative Group; NSDA: National Spasmodic Dysphonia Association; N/A: not available

**Table 2 toxins-13-00840-t002:** Diagnostic criteria for spasmodic dysphonia. (Data from Hyodo M, Auris Nasus Larynx, published by Elsevier, 2021. [6]).

**Requirements (All Mandatory).**
(1) Voice symptoms persist for ≥6 months;
(2) No organic lesion or paralysis of the phonatory organs;
(3) No apparent abnormality in laryngeal function in terms of breathing or swallowing;
(4) No apparent physical or psychological cause prior to disease onset;
(5) No neurological or muscular disease except dystonia.
**1. Main Symptoms:** Symptoms that appear during speech in conjunction with a normal voice.
**ADSD**
(1) Involuntary and intermittent strained or strangled voice;
(2) Involuntary and intermittent voice breaks;
(3) Aperiodic voice tremor;
(4) Effortful phonation.
**ABSD**
(1) Involuntary and intermittent breathy hoarseness;
(2) Involuntary and intermittent aphonia;
(3) Involuntary and intermittent falsetto voice;
(4) Voiceless phonation.
**Mixed Type**
Combination of the symptoms of ADSD and ABSD
**2. Accompanying Symptoms**
(1) Certain words are difficult to pronounce (e.g., words that begin with a vowel by ADSD patients, unvoiced consonants by ABSD patients);
(2) Voice symptoms are reduced or disappear when the voice is high-pitched;
(3) Voice symptoms are reduced or disappear when laughing, crying, whispering, or singing;
(4) Voice symptoms worsen in strained or stressful situations, such as talking on the telephone or during business discussions.
**3. Findings during Phonation**
(1) Laryngoscopic findings
Involuntary and intermittent adduction/abduction of the vocal folds that are synchronized with the voice symptoms.
(2) Findings other than vocal fold findings
Involuntary (unusual) descent or elevation of the larynx, or an abnormal cervical position.
(3) The sensory trick
Voice symptoms are alleviated by touching the neck with a hand, when chewing gum, when tilting the neck, or on topical anesthesia of the laryngeal mucosa.
**4. Treatment Response**
(1) Trial injection of BT into the thyroarytenoid/posterior cricoarytenoid muscle improves the major symptoms;
(2) Systematic voice therapy does not completely resolve the symptoms.
**5. Differential Diagnoses**
(1) Essential or secondary voice tremor;
(2) Muscle tension dysphonia;
(3) Psychogenic dysphonia;
(4) Stuttering.
**<Definite>**
(1) ≥3 main symptoms and all of “5. Differential diagnoses” lacking;
(2) ≥3 main symptoms and ≥3 items of “2. Accompanying symptoms” or “3. Findings during phonation”.
**<Possible>**
(1) ≥3 main symptoms, but at least one of “5. Differential diagnoses” is possible;
(2) 2 main symptoms and ≥2 items of any of “2. Accompanying symptoms”, “3. Findings during phonation” or “4. Treatment response”.

ADSD: adductor spasmodic dysphonia, ABSD: abductor spasmodic dysphonia, BT: botulinum toxin.

**Table 3 toxins-13-00840-t003:** Severity grading of spasmodic dysphonia.

**Subjective grading**	
**1.Voice Handicap Index (VHI)**	Score
(1) 0–24	0
(2) 25–49	1
(3) 50–74	2
(4) 75–120	3
**2. Socio-psychological impairment**	
(1) Normal social life without difficulty in daily conversation;	0
(2) Normal social life with mild difficulty in daily conversation;	1
(3) Moderate impairment of social life because of difficulty in daily conversation, such as difficulty when talking on the telephone or in business discussions;	2
(4) Apparent impairment of social life because of difficulty in daily conversation, such as avoiding talking or socializing with others, quitting a job, becoming fired, or forsaking employment	3
**Objective severity grade**	
Physicians listen to free talk or recitations of standardized sentences.	
(1) Smooth and clear in conversation and recitation;	0
(2) Mild difficulty in conversation or recitation;	1
(3) Moderate difficulty in conversation and recitation, and sometimes hard to hear;	2
(4) Severe difficulty in conversation and recitation, and often very hard to hear.	3
**Overall disease severity** 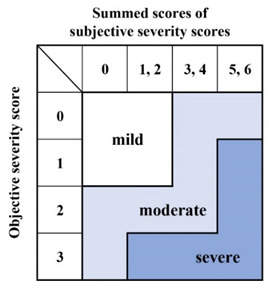	

**Table 4 toxins-13-00840-t004:** Change values of aberrant morae, GRBAS, VHI, and VAS after BT therapy for ADSD. (Data from Hirose K, Laryngoscope Invest Otolaryngol, published by Wiley, 2021. [45]).

			(Mean ± SE)							
		Baseline	Change Value							
			2 w		4 w		8 w		12 w	
number of aberrant morae	BT	19.2 ± 1.36	−9.9 ± 2.66	** #	−7.0 ± 2.30	* #	−6.3 ± 1.90	* #	−3.5 ± 1.42	* #
	Placebo	21.3 ± 1.86	−1.1 ± 0.68		−0.2 ± 0.46		−0.3 ± 0.62		0.4 ± 0.43	
(S) in GRBAS	BT	2.1 ± 0.21	−1.18 ± 0.33	** #	−0.91 ± 0.37	*	−0.73 ± 0.36	*	−0.36 ± 0.24	*
	Placebo	1.9 ± 0.34	−0.18 ± 0.18		−0.27 ± 0.36		−0.00 ± 0.19		−0.27 ± 0.30	
VHI	BT	78.5 ± 5.69	−14.6 ± 7.35		−24.0 ± 9.63	*	−20.6 ± 9.91	*	−16.7 ± 7.59	*
	Placebo	72.5 ± 5.01	−9.8 ± 3.32		−5.3 ± 3.43		−8.0 ± 3.52		−5.7 ± 4.90	
VAS	BT	71.9 ± 5.39	−11.6 ± 8.67		−20.5 ± 8.74		−18.6 ± 10.53		−15.6 ± 8.68	
	Placebo	72.9 ± 5.45	−2.0 ± 4.09		−6.2 ± 4.67		−0.2 ± 4.70		−3.2 ± 3.95	

VHI: Voice Handicap Index, VAS: Visual analogue scale, BT: Botulinum toxin. *: <0.05, **: <0.01 (vs. baseline).; #: <0.05 (BT vs. Placebo).

## Data Availability

The data that support the findings of this study are available on reasonable request from the corresponding author. The data are not publicly available due to privacy or ethical restrictions.

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
