# Peer review of "Botulinum Toxin Therapy: A Series of Clinical Studies on Patients with Spasmodic Dysphonia in Japan"

_toxins, 2021, doi:10.3390/toxins13120840_

Round 1
Reviewer 1 Report
The paper “Botulinum toxin therapy: a series of clinical studies on patients with spasmodic dystonia”, explores the Spasmodic dysphonia (SD), a rare voice disorder caused by involuntary and intermittent 4 spasms of the laryngeal muscles. Authors created diagnostic criteria for SD and a severity grading system to facilitate and homogenize the diagnosis of this disease following Ludlow et al. who proposed a three-step procedure that consists of a screening questionnaire (possible SD), clinical speech examination (probable SD), and laryngeal examination (definite SD). The procedure is concise and clinically useful, although it requires further validation. Our criteria are generally consistent with this approach and convenient for use in the clinic.
The paper is interesting and seeks to shed light on a rare and poorly understood disease
There are some concerns:
- Page 2-3. Epidemiological studies show a wide proportion of this disease from 0.5 to 13cases/100000. There are significant differences between Europe and USA or Japan date. Please discuss this point.
- Page 7 Lines 190 and following. How was performed the Laryngeal endoscopy and what were results of this examination?
- How did they assess whether the BT injections were capable of paralyzing the target muscles?
- Table 4 page 9. Please insert also baseline values for treated and placebo
- Table 4 . Please insert also a comparison for every time between Placebo and BT to complete the results.
- Page 9 line 297: correct Figure 3 in Figure 1
- Page 9 >Please compare AEs data with placebo data.
- Page 9 line 299. Why not curative? In my opinion BT therapy seems to be a not definitive therapy, but also surgery seem to be not always definitive therapy.
Reviewer 2 Report
Spasmodic dysphonia is a rare disease and its diagnosis and treatment selection are not well described. This paper summarizes what has been done through a series of clinical studies including nationwide survey, creation of diagnostic criteria, and randomized and double-blinded clinical trial to facilitate early diagnosis and appropriate therapy. The attempts are appreciated and the paper reads well for a review of the studies. Please address these points:
Since it is in Japan, can this appears in the title?
Do the authors recommend similar criteria for diagnosis and treatment strategy based on the findings that it can be generalized for use in other parts of the globe?
Author Response
Thank you for your peer review and valuable comments.
1. Since it is in Japan, can this appears in the title?
We added "in Japan" in the title.
2. Do the authors recommend similar criteria for diagnosis and treatment strategy based on the findings that it can be generalized for use in other parts of the globe?
Yes we do. We added a comment in the conclusion section (line 328-330).